# Recent Trends of Foaming in Polymer Processing: A Review

**DOI:** 10.3390/polym11060953

**Published:** 2019-06-01

**Authors:** Fan-Long Jin, Miao Zhao, Mira Park, Soo-Jin Park

**Affiliations:** 1Department of Polymer Materials, Jilin Institute of Chemical Technology, Jilin 132022, China; jinfanlong@163.com (F.-L.J.); namezhaomiao@sina.com (M.Z.); 2Department of Organic Materials & Fiber Engineering, Chonbuk National University, Jeonju 54896, Korea; 3Department of Chemistry, Inha University, Nam-gu, Incheon 402-751, Korea

**Keywords:** polymer, foam, foaming process, blowing agent, foaming method

## Abstract

Polymer foams have low density, good heat insulation, good sound insulation effects, high specific strength, and high corrosion resistance, and are widely used in civil and industrial applications. In this paper, the classification of polymer foams, principles of the foaming process, types of blowing agents, and raw materials of polymer foams are reviewed. The research progress of various foaming methods and the current problems and possible solutions are discussed in detail.

## 1. Introduction

Polymer foam is an important polymer material whose polymer matrix contains a large number of tiny foam holes inside and is also known as a porous polymer material. Compared to bulky polymer materials, polymer foam has many advantages, such as low density, good heat insulation, good sound insulation effects, high specific strength, and resistance to corrosion. At present, polymer foam is one of the most widely used polymer materials and plays a very important role in the polymer industry [1,2,3,4,5,6,7,8].

Polymer foam was prepared from polymer using mechanical, physical, and chemical foaming. The processing techniques of polymer foams mainly include foam extrusion molding and foam injection molding. These molding methods with a supercritical fluid, such as carbon dioxide (CO_2_) or nitrogen (N_2_), can be foamed with lighter weight and higher-dimensional stability of polymer foams than their solid counterparts [9,10,11,12].

Polymer foams are widely used in several industrial applications. For example, polyurethane (PU) soft foam can be used in furniture, composite fabrics, clothing, shoes and hats, car cushions, and sports equipment [13], while PU hard foam can be used in the production of refrigerators, freezers, and refrigerated containers [14,15]. Polypropylene (PP) foam is widely used in daily necessities, military industries, transportation, and aerospace, such as in the production of various seat cushions, thermal insulation materials, shockproof materials, packaging materials, and building materials [16,17,18]. Phenolic foam is mainly used in the fields of architecture, automobiles, electrical and electronic applications, the iron and steel industry, and aerospace [19,20,21,22]. It can meet the requirements of heat insulation and combustion-resistant building materials for large-scale sports and entertainment venues, high-rise buildings, and high-speed vehicles. Polyimide (PI) foam, with high thermal insulation, great thermal stability, and high fire resistance, has broad prospects for application and development in the aerospace, aircraft, and marine fields [23,24,25,26,27,28].

With the ongoing development of polymer foams, they have been widely used in various fields. As the most important of industrial polymer products, polymer foams will have a broad space for development and prospects. This paper mainly introduces polymer foaming with the research progress of various foaming methods and current problems and possible solutions discussed in detail.

## 2. Polymer Foaming and Processing

### 2.1. Classification of Polymer Foams

Polymer foams can be described in terms of their density, average cell size, and cell density. There are a variety of classification methods for polymer foams. Among them, a common classification method involves the classification of polymer foams into three categories hardness, density, and foaming structure.

#### 2.1.1. Hardness Classification

Under temperature conditions of 23 °C and relative humidity of 50%, polymer foams are mainly classified into three categories, based on the size of the elastic modulus. If the elastic modulus of a polymer foam is less than 68.6 MPa, the polymer foam is called a soft polymer foam, e.g., soft polyvinyl chloride (PVC), polyvinyl formal (PVF), and PP foams [29,30,31]. If the elastic modulus of a polymer foam is 68.6−686 MPa, the polymer foam is called a semi-rigid polymer foam, e.g., semi-hard PU and PI foams [32,33]. If the elastic modulus of a polymer foam is greater than 686 MPa, the polymer foam is called a rigid polymer foam, e.g., hard PU, polystyrene (PS), and phenolic foams [34,35,36].

#### 2.1.2. Density Classification

Polymer foams are classified into three categories based on density. If the density of a polymer foam is less than 0.1 g/m^3^, the polymer foam is considered high-foaming, with examples such as high-foaming polyethylene (PE) and PP foams [37,38,39]. If the density of a polymer foam is 0.1–0.4 g/m^3^, the polymer foam is considered medium-foaming, with medium-foaming PI foam serving as an example [40]. If the density of a polymer foam is greater than 0.4 g/m^3^, the polymer foam is considered low-foaming, with low-foaming PP foam as an example [41].

#### 2.1.3. Classification of Bubble Structure

Polymer foams are classified into two categories based on the structure of the cell hole. One category is polymer foam with open cells connected to each other. Examples include melamine-formaldehyde foam and epoxy foam [42,43,44,45]. The other category is closed-cell foam with separate cells, in which PS foam, polymethylacrylimide foam, and PE foam are examples [46,47].

### 2.2. Blowing Agents

Blowing agents can be separated into physical blowing agents and chemical blowing agents based on the method of gas production during the foaming process.

#### 2.2.1. Physical Blowing Agent

Physical blowing agents can be divided into two categories, i.e., inorganic and organic blowing agents. Inorganic blowing agents contain nitrogen, carbon dioxide, water, and air [48,49,50,51]. Organic blowing agents include pentane, hexane, dichloroethane, and Freon [52].

#### 2.2.2. Chemical Blowing Agent

Chemical blowing agents are classified into two categories, i.e., inorganic and organic foaming agents. The blowing agents can also be separated into two types, thermal decomposition, and reaction blowing agents.

The inorganic blowing agent is mainly used in synthetic rubber, natural rubber, and rubber foam products. An inorganic thermal decomposition blowing agent mainly includes bicarbonate, carbonate, and nitrite. An inorganic reactive blowing agent contains sodium bicarbonate or zinc powder, an acid reaction, hydrogen peroxide, and a yeast reaction [53,54].

The organic blowing agent has advantages such as good dispersibility, stable gas output, and uniform bubbles. The organic reactive blowing agent has isocyanate compounds. The organic thermal decomposition blowing agents include azo foaming agent, nitroso foaming agent, and acylhydrazide foaming agent [55,56,57,58].

#### 2.2.3. Expandable Bead

Expandable bead is a core-shell structure. The shell is a thermoplastic acrylic resin polymer, and the core is a hollow spherical microparticle composed of alkanes. The expandable bead serves to provide foaming through the expansion of the microsphere. When heated, the volume of the expandable bead can be rapidly expanded by dozens of times, achieving an excellent foaming effect, such as with expandable PE, PP, and PS [59].

### 2.3. Principle of Foaming Process

The foaming process of polymers can be separated into three stages: Cell formation, cell growth, and cell stabilization. Figure 1 shows a schematic diagram of the foaming process.

#### 2.3.1. Cell Formation

A blowing agent (or gas) is added to a molten polymer under certain conditions, and then, a large amount of gas is produced through a series of chemical reactions (or added gas), which forms a polymer/gas solution. When the amount of the gas gradually increases, the solution becomes supersaturated, and the gas escapes from the solution. The escaped gas forms the cell nucleus through nucleation [12,60,61].

#### 2.3.2. Cell Growth

After the formation of cells, the pressure of the gas inside the cell is inversely proportional to the radius of the cell. Thus, the smaller the cell, the greater the pressure inside the cell. When two cells of different sizes are close to each other, the gas will spread from small cell to a large cell, and the two cells merge together. As a result of nucleation, the number of cells increases, and the diameter of the cell hole expands. Thus, the cell can grow [62,63].

#### 2.3.3. Cell Stabilization

On account of the formation and growth of a large number of cells, the surface area and volume of the foam system continuously increases, and the cell wall becomes thinner. Thus, the foam system becomes unstable. Cells are usually stabilized by cooling or by adding surfactants [64,65].

### 2.4. Foaming Methods

Foaming methods have various classification methods. In this review, foaming methods are categorized as mechanical foaming, physical foaming, and chemical foaming.

#### 2.4.1. Mechanical Foaming

Figure 2 shows a schematic diagram of mechanical foaming. Air is added in polymer resins by mechanical stirring, and then, the resin is foamed.

Advantages of mechanical foaming: (1) No additional foaming agent, simple process equipment, and easy to operate, (2) green environmental protection, non-toxic, and safe, (3) low cost and high efficiency.

Disadvantages of mechanical foaming: Higher requirements on equipment.

#### 2.4.2. Physical Foaming

In physical foaming, a low-boiling-point (BP) liquid and a polymer are blended and then foamed through pressuring and heating [66], as shown in Figure 3.

Advantages of physical foaming: (1) The cost of physical foaming agent is relative low, especially carbon dioxide and nitrogen, (2) the foaming method is pollution-free, which has a high application value, (3) this method has no residue after foaming and has little influence on the properties of foaming plastics.

Disadvantages of physical foaming: (1) Needs special injection molding machine, and auxiliary equipment, (2) needs high technical requirements.

#### 2.4.3. Chemical Foaming

Figure 4 shows a schematic diagram of chemical foaming. The chemical foaming process can be undertaken by two methods. The first method is as follows: A blowing agent is added to a molten polymer and is decomposed to release gas. Then, the polymer is foamed by pressuring and heating, as shown in Figure 4a. The second method is shown in Figure 4b, in which chemical reactions between the two polymers occur to produce inert gases, and then, the polymers are foamed.

Advantages of chemical foaming: (1) Chemical foaming agent can be thermally decomposed in a specific temperature range and released one or more gases, thus it can be suitable for polymer resins that have melt viscosity in a specific temperature range, (2) chemical foaming can be carried out with ordinary injection molding machine.

Disadvantages of chemical foaming: High mold manufacturing precision is required, the mold cost is high, and a second clamping pressure device is needed during high-pressure foaming process.

### 2.5. Processing Techniques

#### 2.5.1. Foam Extrusion Molding

We take the co-rotational twin-screw extruder as an example to explain foam extrusion molding. A supercritical fluid, such as carbon dioxide (CO_2_) or nitrogen (N_2_), was used as a blowing agent. The extruder consists of several heating zones. The extruder was heated to a specified temperature, and the discharge rate and screw speed were set to specified values according to the process conditions. A polymer (polymer blends or polymer composite) was added in the hopper of the extruder and extruded under the processing conditions. A supercritical fluid was introduced into a zone of the barrel using a syringe pump. Supercritical fluid was dissolved in the polymer melt, reducing the melt viscosity of the polymer. By reducing the foaming temperature, the cell density increased, and the melt strength increased. Finally, microcellular foams were obtained [11,67,68].

#### 2.5.2. Foam Injection Molding

We take the injection molding machine as an example to explain foam injection molding. The injection machine consists of several injection temperatures from the hopper to the nozzle. The injection molding parameters, such as injection temperature, injection pressure, injection speed, mold temperature, holding time, and cooling time were set to the fixed values according to the processing conditions. A polymer (polymer blends or polymer composite) was plasticized under the processing conditions. A supercritical fluid was metered into the barrel. The supercritical fluid was quickly dissolved into the polymer melt to form a homogeneous polymer/gas solution. The solution was injected into the mold cavity suffering a rapid pressure drop, which could simultaneously produce homogeneous and/or heterogeneous cell nucleation and cell growth to foaming. The visual appearance and characteristics of the injected foams are highly dependent on the raw material, mold design, and processing conditions [69,70].

### 2.6. Raw Materials of Polymer Foams

The raw materials of polymer foams can be categorized as plastics and rubbers. The plastic is further categorized as thermosetting resins and thermoplastics.

#### 2.6.1. Plastics

##### Thermosetting Resins

Thermosetting resins generally exist in liquid form and are cured through a chemical reaction. The raw materials of thermosetting resins have epoxy resin, phenolic resin, polyester, and PU [71,72,73,74,75,76,77,78].

Esmaeili et al. [72] used epoxy resin as a raw material to prepare bio-based thermosetting epoxy foam for dye decontamination and thermo-protecting applications. Li et al. [74] presented new foaming formulations for the production of bio-phenol formaldehyde foams. Bakir et al. [77] used aromatic thermosetting copolyester as a raw material to prepare lightweight structural foams with high thermal- and mechanical-performance.

##### Thermoplastics

At a certain temperature, thermoplastics can melt or soften into any shape, and the shape does not change after cooling. The thermoplastics for polymer foams include PE, PP, PS, PVC, and poly (methyl methacrylate) (PMMA) [79,80,81,82,83,84,85,86,87,88,89].

Wu et al. [79] used low-density PE as a raw material to study cell growth in low-density PE foaming processes. Wang et al. [81] prepared microcellular-insert injection-molded PP composite foams. Tang et al. [84] analyzed the mechanical performance of PS foam. Senol et al. [87] prepared closed-cell PVC foams through 3D DIC technique. Notario et al. [89] studied the dielectric behavior of porous PMMA.

#### 2.6.2. Rubbers

Rubbers, as the raw materials for polymer foams, include natural rubber, silicone rubber, ethylene-propylene diene monomer (EPDM) rubber, and styrene butadiene rubber [90,91,92,93,94,95,96].

Huang et al. [90] investigated the effects of bottom ventilation on the fire behavior of natural rubber latex foam. Bai et al. [92] prepared microcellular silicone rubber/nanographite foam with enhanced mechanical performance. Liu et al. [94] fabricated superhydrophobic and superoleophilic-modified EPDM foam rubber by a facile approach for oil/water separation. Ji et al. [96] prepared styrene butadiene rubber/ethylene vinyl acetate composite foams with skeleton support structure based on an alternately cross-linking process.

## 3. Research Progress of Various Foaming Methods

Whether mechanical foaming, physical foaming or chemical foaming, the common feature is polymer resin must be in a liquid state or a plastic state with low viscosity. Three foaming methods are not necessarily suitable for the same kind of polymer resin due to the different requirements of foaming plastics for different purposes and the different properties of each resin.

Compared with other foaming methods, mechanical foaming is easy to control, green, non-toxic, and safe. The degree of foaming is well controlled by physical foaming, which is relatively stable. The degree of foaming by chemical foaming is unstable. There are residues after chemical foaming, but the residues do not affect subsequent production, storage, and use.

Mechanical foaming is commonly used in epoxy resin, phenolic resin, PP, and urea-formaldehyde resin. Physical foaming is suitable for a wide variety of plastics, such as PP, PMMA, PLA, and phenolic resin. Chemical foaming is often used in the production of PE, PP, PVC, PI, polyamide 6 (PA6), and PU foams.

### 3.1. Mechanical Foaming

Song et al. [52] prepared macroporous bio-based epoxy resin by mechanical frothing. Their results indicated that heating the air-in-resin liquid foams prior to their gelation decreased the viscosity of the resin mixture and increased the pressure of the air bubbles, leading to an isotropic expansion of the air bubbles. Correspondingly, the porosity of the resulting macroporous polymers increased from 71% to 85%. Mao et al. [18] studied the effects of nano-CaCO_3_ content on the crystallization, mechanical properties, and cell structure of PP foams prepared using supercritical nitrogen as a physical foaming agent. Their results indicated that using the nano-CaCO_3_ as a bubble-nucleating agent improves the cell structure of PP.

Li et al. [19] prepared cork-powder-reinforced larch tannin-based rigid phenolic foams. Their results indicated that the reinforced phenolic foams showed excellent cell morphology, high compressive strength, low thermal conductivity, low friability, and good thermal properties as compared with unreinforced phenolic foams. Alander et al. [36] prepared rigid and soft protein biofoams by high-speed stirring of pristine wheat gluten powder in water at room temperature, followed by drying. Experimental results indicated that the foams with medium to high density readily absorbed both hydrophobic and hydrophilic liquids. The compression strength of the foam was improved by making a denser foam.

Liu et al. [61] fabricated an intrinsic flame-retardant urea-formaldehyde/aramid fiber composite foam via an in-situ water-based foaming process. Their results indicated that the compressive strength and modulus of the foam increased by 198% and 257%, respectively, when the content of aramid fiber was 2 wt%. Jin et al. [97] studied the physico-mechanical and fire properties of PU/melamine-formaldehyde (MF)-interpenetrating polymer network foams prepared using mechanical foaming. Experimental results indicated that the density of the PU foams increased as the MF content increased, as shown in Figure 5. SEM micrographs revealed that the average pore size of the foam initially decreased upon the addition of MF and then increased as more MF was added, as shown in Figure 6.

### 3.2. Physical Foaming

Zhou et al. [10] prepared PMMA microporous foams with a multi-layer cell structure via a combination of multi-layer hot melt pressing and a supercritical carbon dioxide foaming method. Their results indicated that nucleation and the directional growth of the cells are promoted by the introduction of the multi-layer interface into the PMMA matrix. The compression strength of the multi-layer foam in the horizontal (20.27 MPa) and vertical loading directions (11.84 MPa) is much higher than that of the uniform foam.

Wang et al. [11] prepared lightweight and tough nanocellular PP/polytetrafluoroethylene (PTFE) nanocomposite foams with defect-free surfaces obtained using in situ nanofibrillation and nanocellular injection molding. Their results indicated that the nanocellular PP/PTFE nanocomposite foams show significantly enhanced mechanical properties as compared to the regular PP foam. The nanocellular foam exhibited an outstanding surface appearance compared with that of regular foam. Ge et al. [21] prepared 3-pentadecyl-phenol-modified foamable phenolic resin. Experimental results indicated that the modified phenolic foams exhibit a more regular and dense network structure. The compressive strength (Figure 7), bending strength, and limiting oxygen index (LOI) of the foams were improved, and the water absorption rate was decreased by the addition of 3-pentadecyl-phenol.

Pradeep et al. [98] studied physical, and compatibilized blends of PLA/poly (butylene succinate-co-adipate) (PBSA) processed via supercritical fluid-assisted injection molding technology, using nitrogen as a facile physical blowing agent. Their results indicated that the PLA/PBSA foams reveal increased crystallinity and storage modulus as compared to their physically foamed counterparts. Gazzani et al. [99] prepared a high-temperature epoxy foam. Table 1 shows the density of epoxy foam samples as a function of temperature. The density of the samples decreased with increasing temperature. The resulting epoxy foam has a glass transition temperature of 270 °C.

Ellingham et al. [68] prepared PP/graphene nanocomposites using a twin-screw extruder equipped with a simple CO_2_ injection unit. Their results indicated that the addition of CO_2_ into the melt at pressures below the supercritical point allowed for foaming to occur within the barrel of the extruder and upon exiting from the die. Wang et al. [3] fabricated lightweight and strong microcellular PP/talcum nanocomposite foam using twin-screw compounding. Their results indicated that the PP/talcum nanocomposite foam shows simultaneously improved strength, rigidity, and toughness. Albooyeh et al. [83] investigated the effect of multiwall carbon nanotubes (MWCNTs) on the vibration and morphological properties of short glass fiber (SGF)-reinforced PP composite foams prepared using melt-compounding in a twin-screw extruder. The experimental results showed that the addition of SGF particles to the PP foam would significantly increase the natural frequencies. The results also indicated that the damping factor and natural frequencies of the SGF/PP foams increased with the addition of MWCNTs. Keshtkar et al. [67] investigated the effects of nanoclays on the foamability of polylactide (PLA) in continuous extrusion foaming, using supercritical CO_2_ as a blowing agent. The results showed that both the cell density and the expansion ratio were greatly promoted with increased clay content and dissolved CO_2_, as well as by the possibility of nucleated crystals.

Wang et al. [11] reported lightweight and tough nanocellular PP/PTFE nanocomposite foams with defect-free surfaces, prepared using in situ nanofibrillation and nanocellular injection molding. Their results indicated that the impact strength of the nanocellular foam was 700% higher than that of the regular foam and 200% higher than that of the unfoamed product. The nanocellular PP/PTFE nanocomposite foam showed outstanding surface appearance without any silver or swirl marks compared to regular foam. Mao et al. [18] studied the effects of nano-CaCO_3_ content on the crystallization, mechanical properties, and cell structure of PP nanocomposites in microcellular injection molding using supercritical nitrogen as a physical foaming agent. The results showed that the addition of nano-CaCO_3_ to PP could improve its mechanical properties and cell structure. The thermal stability and crystallinity of the nanocomposites were enhanced with increasing nano-CaCO_3_ content. Zhao et al. [4] fabricated high thermal insulation and compressive strength PP foams by high-pressure foam injection molding and mold opening of nano-fibrillar composites. Their results indicated that the PP/PTFE foams show significantly improved thermal insulation performance and unique cell wall structures with micro-holes and/or nano-fibrils over PP foams. The results also demonstrated that smaller cell size leads to improved compressive strength.

Wang et al. [81] prepared all-PP composite foams by inserting injection molding combined with microcellular injection molding. Their results indicated that the all-PP composite foams produced at the injection temperature of 220 °C and injection speed of 40 mm/s show 4.23% weight reduction, 16.52% increase in flexural strength, and 20% increase in flexural modulus when compared with solid PP. Pradeep et al. [98] investigated the thermal and thermomechanical properties of biodegradable PLA/PBSA composites processed via supercritical fluid-assisted foam injection molding. Their results indicated that the addition of triphenyl phosphite (TPP) led to an increase in molecular weight and a shift in melting temperature. The compatible PLA/PBSA foams have increased crystallinity and storage modulus compared to their physically prepared foams.

### 3.3. Chemical Foaming

Sadik et al. [53] investigated the decomposition kinetics of citric acid and sodium bicarbonate, and their combinations in masterbatches with low-density PE, in view of their use in injection polymer foaming. Their results indicated that for sodium bicarbonate compounds, the kinetics can be modeled by first-order equations, while for citric acid compounds, two steps of decomposition are indicated. Wang et al. [3] prepared porous polyamide 6 (PA6) materials by a facile solution-foaming strategy. Experimental results indicated that sodium carbonate aqueous solution acted as the foaming agent that reacted with formic acid, generating CO_2_ and causing the phase separation of PA. The resulting porous PA materials exhibited low thermal conductivity, high crystallinity, and good mechanical properties (Table 2). Ramya et al. [24] synthesized novel regenerable 13X zeolite-PI adsorbent foams. Their results indicated that isocyanate reacts with distilled water to produce CO_2_ and a primary amine to create pores in the foam during the blowing reaction. The PI foams showed superior CO_2_/N_2_ selectivity when compared to other adsorbents in the literature.

Long et al. [58] prepared PU foams using polypropylene glycol (PPG)-grafted polyethyleneimines (PEIs) as blowing agents. Their results indicated that the reaction extent of the restored amines decreased by increasing the chain length on the PPG side or in the grafting rate. The macromolecular mobility decreased by increasing the PEI backbone molecular weight, which caused a decrease in the reaction extent as well. Liu et al. [57] prepared green PU foams via renewable castor-oil-derived polyol and carbon-dioxide releasing blowing agents from alkylated PEIs. Experimental results indicated that the alkylation of the branched PEI amine groups enhances the dispersibility of the resultant CO_2_ adducts into the PU foaming mixtures containing the castor-oil-derived polyol.

Gong et al. [9] prepared PP composite foams by chemical foaming technology. Their results indicated that the foaming quality was significantly improved after the introduction of thermoplastic rubber and polyolefin elastomer. The impact properties of the PP composite foams were greater than that of the unfoamed sample at temperatures between −80 and −20 °C. Li et al. [26] fabricated a series of reinforced PI/carbon fiber (CF) composite foams through thermal foaming. Experimental results revealed that the chopped CF acted as a nucleation agent in the foaming process. The mechanical properties of PI foams were significantly improved owing to the incorporation of chopped CF.

Sun et al. [27] studied the effects of aramid honeycomb core (ARHC) on the mechanical properties of isocyanate-based PI foams. The mechanical property tests showed that the compression strength of PI foams increased with the addition of ARHC. Laguna-Gutierrez et al. [37] investigated the mechanical properties and effective diffusion coefficient under static creep loading of low-density foams based on PE/clay nanocomposites. Their results indicated that the polymer crystallinity increased and the polymer degradation window broadened with the addition of clays. The tensile and compression properties of the foams improved with the addition of clays.

Mantaranon et al. [100] prepared polyoxymethylene (POM) foam based on using a blowing agent, i.e., azodicarbonamide (ADAC). Their results indicated that the ADCA initiates cyanic acid with CO_2_, and allows for monodispersed bubbles without significant degradation of POM. The cell size increases from 80 mm to 150 mm, with an increase in ADCA content from 1 phr to 3 phr, as shown in Figure 8. Wu et al. [55] presented a new core-back foam injection molding method with chemical blowing agents. Experimental results indicated that a closed shell composed of dense polymer skins can be created right before melt filling by combining core-back foam injection and secondary filling. The mechanical testing results showed that new technology can produce polymer foam with simultaneously enhanced tensile strength, elastic modulus, and notch impact strength. Michałowski et al. [56] prepared porous PVC with reduced apparent density and good mechanical properties, using ADAC as a blowing agent. Their results indicated that the structures of PVC foams were improved and the apparent density decreased with the addition of carbon black. The apparent density of PVC foams also decreased when montmorillonite was added, as shown in Table 3.

Wu et al. [55] presented a new core-back chemical foam injection molding method that has a unique secondary filling stage right after the core-back operation. The mechanical testing results showed that the new technology can produce plastic foam with simultaneously enhanced tensile strength, elastic modulus, and notch impact strength. This technology can also improve the surface appearance of the foamed sample.

## 4. Current Problems and Possible Solutions

### 4.1. Current Problems

(1) With the growing use of petroleum-based polymer foams, the global accumulation of plastic waste has increased continuously every year. For example, packing foams prepared from petroleum-based polymers are used to protect equipment or items from damage during transportation. However, after the foams were used, they were discarded, becoming white trash. In addition, buried polymer foams reside underground for hundreds of years without decay, causing serious pollution to the environment [101,102,103,104,105,106].

(2) Most polymer foams are highly flammable and exhibit poor flame retardation. For example, PS foams, as the most commonly used form of commercial thermal and acoustic insulation have been widely used in building thermal insulation. However, they often produced heavy smoke and release toxic gases during burning, which seriously threatens the lives of humans and limits their further application [107,108,109,110].

(3) Most rigid polymer foams such as rigid PU, PVC, epoxy, and phenolic foams, are brittle and have poor fracture resistance [111,112,113,114]. For example, packaging foams of electrical appliances were broken during transportation. In severe cases, electrical appliances are also damaged.

### 4.2. Possible Solutions

#### 4.2.1. Environmental Pollution

The environmental pollution caused by petroleum-based polymer foams can be solved by the following two methods. One method is preparing biodegradable polymer foams such as poly(butylene succinate) (PBS) foam, bio-based PU foam, polycaprolactone (PCL) foam, and poly(lactic acid) (PLA) foam [115,116,117,118,119,120,121]. Yue et al. [115] prepared sustainable PBS foams invoking a heat-counteracted strategy. The closed cell fraction of PBS foams increased from 21% to 99% with the increase in the endothermic expandable microsphere (EnEMs)/AC loading ratio, as shown in Figure 9. The flexural strength of PBS foams was increased from 3 MPa to 21 MPa. In addition, the thermal conductivity of the foams decreased from 58 to 22 mW/(m·K). Önder et al. [117] prepared monolithic PCL foams with controlled morphology. Their results indicated that the PCL foams with pore sizes of 10–450 μm, porosity of 83–91% and various morphologies were obtained by selectively adjusting the process parameters. Önder et al. [118] also fabricated rigid PLA foams via thermally-induced phase separation. Experimental results indicated that PLA foams exhibit high porosities of 85.1–92.8% and pore sizes of 25–400 μm. Członka et al. [120] reinforced soybean-oil-based rigid PU foams (RPUFs) with different weight ratios of chicken feather filler. Their results indicated that the composition foam modified with 0.1 wt% of chicken feathers has a higher compression strength and lower thermal conductivity as compared with unfilled foam.

Another method for addressing environmental pollution is recycling waste polymer foams, such as polyethylene terephthalate (PET) foam and phenolic foam [122,123,124,125,126,127,128]. Gaidukova et al. [124] prepared PU rigid foams from bio-based and recycled PET. The compression tests indicated that the rigidity of PU rigid foams increased almost three-fold, and the strength increased almost two-fold by the addition of glycerol and adipic acid. The water absorption of the foams was below 3 wt%. Lai et al. [125] prepared PET composite foams using recycled PET foams by a thermochemical extrusion foaming process. The resulting PET foams showed a crystallization temperature of 217 °C, flexural strength of 623 kgf/cm^2^, flexural modulus of 22,881 kgf/cm^2^, density of 0.85 g/cm^3^, average cell size of 160 μm, and Izod impact strength of 155 J/m. Modesti et al. [128] studied the recyclability of polyisocyanurate waste with a high isocyanate index. Their results indicated that new rigid polyisocyanurate foams synthesized from recovered polyols showed better mechanical properties and cellular structure as compared with those of foams obtained from virgin polyols.

#### 4.2.2. Flame Retardancy

The flame retardancy of polymer foams can be improved by using the following two methods. One method is adding flame retardants into the foaming formula using physical blending [129,130,131,132,133,134]. Sun et al. [27] studied the effects of ARHC on the flame retardancy of isocyanate-based PI foams. Their results indicated that the flame retardancy of PI foams was significantly increased by the addition of ARHC. Li et al. [19] studied the fire properties of larch tannin-based rigid phenolic foams reinforced with cork powder. Experimental results indicated that the reinforced phenolic foams showed a maximum LOI of 47.8%. Liu et al. [61] investigated the flame retardance of urea-formaldehyde/aramid fiber composite foams. The UL-94 test results showed that the foams were classified with V-0 rating with the addition of aramid fiber.

Wang et al. [129] prepared core-shell expandable graphite (EG) @ aluminum hydroxide (ATH) as a flame-retardant to enhance the flame-retardant performance of rigid PU foams. Their results indicated that the LOI of PU foams increased from 21.5% to 29.6% with the addition of 11.5 wt% EG@ATH. Liu et al. [130] synthesized a magnesium amino-*tris*-(methylenephosphonate)-reduced graphene oxide (Mg-rGO) hybrid to prepare a flame-retardant and toughened phenolic foam. Experimental results indicated that the LOI of the foam with 4 phr Mg-rGO increased to 41.5% compared with 38% for the untreated foam, as shown in Table 4. The peak heat release rate and total heat release of the foam decreased by 28.7% and 18.4%, respectively. Vadas et al. [132] studied flame retardancy of microcellular PLA foams prepared by supercritical CO_2_-assisted extrusion. Their results indicated that the PLA foams showed excellent flame retardancy, a UL-94 V-0 rate and an LOI value of 31.5%, with the addition of 19.5% intumescent flame retardant.

Another method is adding reactive flame retardants into the foaming formula [133,134,135,136,137,138,139,140]. Jin et al. [96] studied the flame retardance of PU/MF interpenetrating polymer network foams. Their results indicated that the linear burn rates of the foams decreased significantly from 236 to 126 mm/s with increasing MF content, as shown in Figure 10. Wang et al. [135] synthesized a reactive flame-retardant triol (TDHTPP), based on a triazine and phosphate structure, to improve the flame-retardance of rigid PU foams. Experimental results indicated that the TDHTPP-incorporated PU foams showed higher compressive strength and lower thermal conductivity than those of the neat rigid PU foams. The PU foams displayed a UL-94 V-0 rating with the addition of 5 wt % TDHTPP. Rao et al. [136] studied the flame-retardant and smoke-suppressant capabilities of flexible PU foams based on reactive phosphorus-containing polyol (PDEO) and expandable graphite (EG). The LOI value of the PU foam with 10 wt% EG and 5 wt% PDEO was 24.5%, which was much higher than those of PU/PDEO and PU/EG. Chen et al. [137] prepared epoxy-resin-modified polyisocyanurate (EP-PIR) foams by the reaction of polymethylene polyphenylene isocyanate (PAPI) and diglycidyl ether of bisphenol-A (DGEBA). The UL94 test results revealed that the foams with a (PAPI)/(DGEBA) ratio above 2.5 could reach V-0 classification. The LOI values of the foams increased linearly from 24.5% to 30.0% with an increase in the (PAPI)/(DGEBA) ratio.

#### 4.2.3. Fracture Toughness

Most rigid polymer foams are brittle and easy to fracture in the preparation and application processes. The fracture toughness of the polymer foams can be improved by the addition of toughening agents [140,141]. The toughening agents can be separated into addition-type (elastomers, inorganic nanoparticles) and reaction-type (polyvinyl alcohol, polyethylene glycol, and PU prepolymer) agents [100,142,143,144,145,146,147,148,149,150,151,152]. Wang et al. [11] studied the fracture toughness of nanocellular PP/PTFE nanocomposite foams. Their results indicated that the impact strength of the nanocellular PP/PTFE nanocomposite foam was 700% higher than that of the regular foam and 200% higher than that of the unfoamed product. Mao et al. [18] investigated the effects of nano-CaCO_3_ content on the impact strength of PP foams. The impact strength of the foams first increased and then decreased with increasing nano-CaCO_3_ content, as shown in Figure 11.

Mantaranon et al. [100] studied the mechanical properties and impact strength of POM foams. Their results indicated that the impact strength of the foams increased from 3.7 to 9.4 kJ/m^2^ with increasing ADCA content from 1 to 3 phr. The impact strength, toughness, and tensile modulus of the foams increased with increasing compression pressure. Liu et al. [150] prepared cellulose-based, ultra-lightweight pulp foams reinforced with microfibrillated cellulose (MFC). Experimental results indicated that the strength of pulp foam was significantly increased by the addition of MFC. The density of the foams was lower than 0.02 g/cm^3^. Ghanbari et al. [151] studied the effect of cellulose nanofibers (CNFs) on the thermal aspects, dynamic mechanical analysis, density, and water uptake of thermoplastic starch (TPS)-foamed composites. Their results indicated that the glass transition temperature (T_g_) of the TPS foams increased with the incorporation of CNFs. The apparent density and water absorption of the foams decreased by the addition of the CNF filler. Członka et al. [152] investigated linseed oil as a natural modifier for rigid PU foams. Experimental results indicated that the PU foams containing linseed oil demonstrate a variety of favorable properties, including improvement in the mechanical strength characteristics.

## 5. Conclusions

Polymer foams as a kind of lightweight material have many superior properties, such as low density, good thermal insulation, excellent sound insulation, high specific strength, and electromagnetic shielding compared to bulky polymer materials, which are widely used in civil and industrial applications, such as insulation, packaging, medical, automotive industries, aircraft, and aerospace fields.

Polymer foams are generally prepared using various foaming techniques, such as batch foaming, bead foaming, extrusion foaming, and injection foam molding. Compared with the regular foaming process using a chemical blowing agent, microcellular foaming with a physical blowing agent can produce polymer foams with much finer cells, which have highly specific properties, including good thermal stability, excellent sound absorption properties, and a low thermal conductivity and dielectric constant. However, there are still many problems in the preparation and application of polymer foams, such as low crystallinity, low recycling of polymer foams, and poor flame retardation. Recent studies have focused on (1) developing supercritical-fluid-assisted injection molding, (2) developing green polymer foams prepared from renewable raw materials, (3) recycling waste of polymer foams, (4) increasing the crystallization rate using nano-fillers as nucleating agents in the foaming of crystalline polymers, (5) improving flame retardancy of polymer foams.

## Figures and Tables

**Figure 1 polymers-11-00953-f001:**
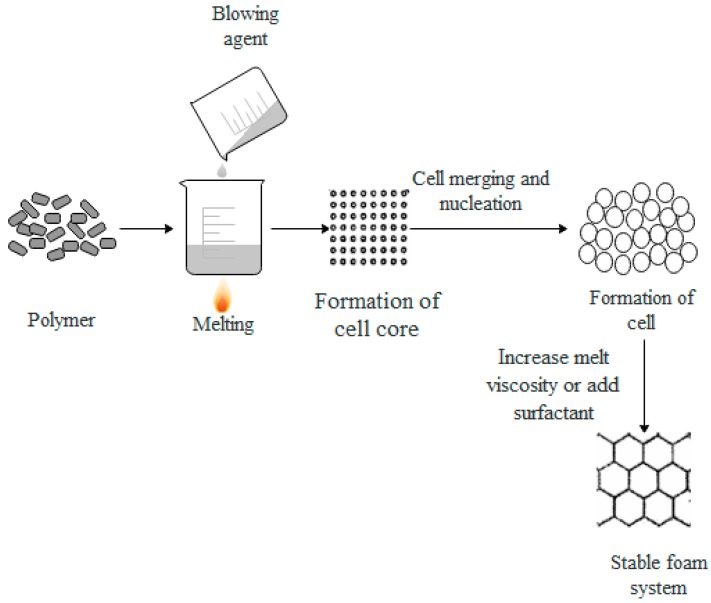
Schematic diagram of the foaming process.

**Figure 2 polymers-11-00953-f002:**
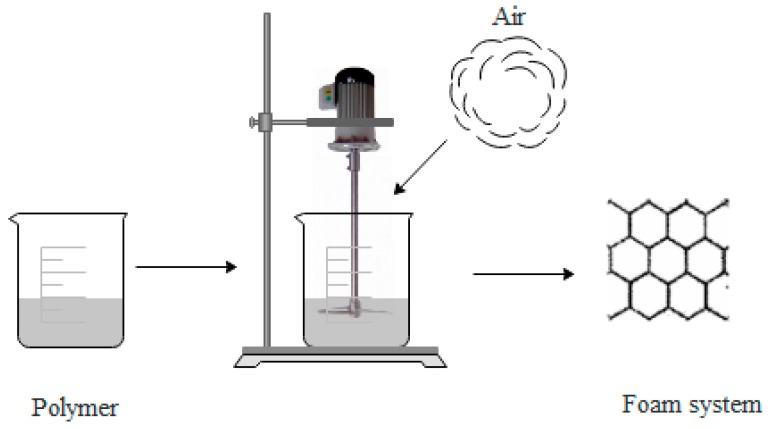
Schematic diagram of mechanical foaming.

**Figure 3 polymers-11-00953-f003:**
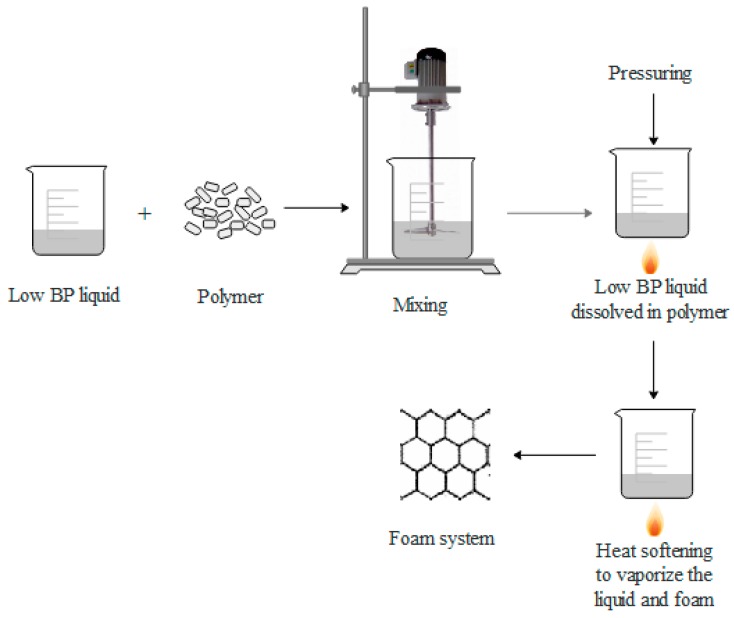
Schematic diagram of physical foaming, reproduced from [66] under open access license.

**Figure 4 polymers-11-00953-f004:**
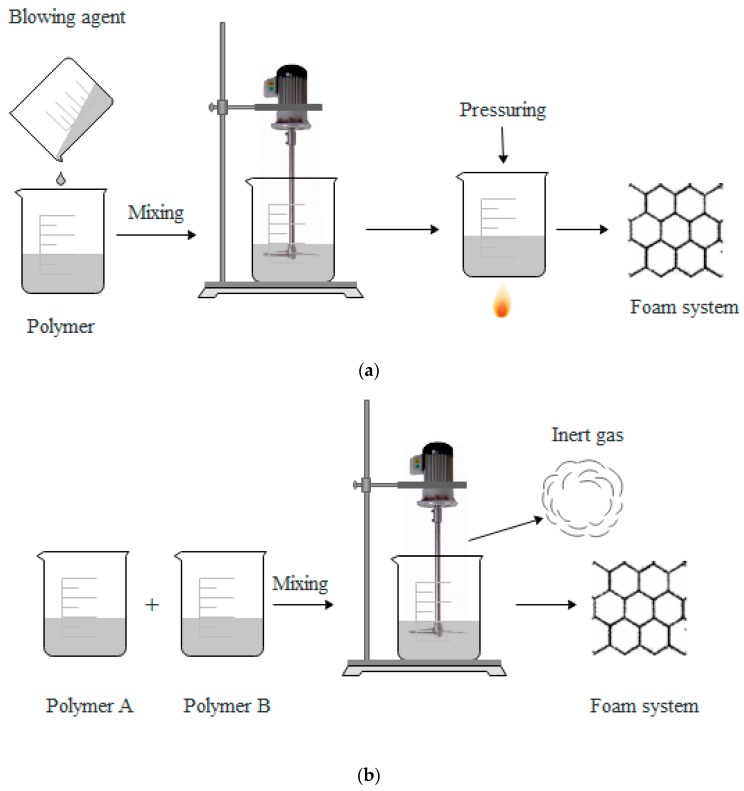
Schematic diagram of chemical foaming.

**Figure 5 polymers-11-00953-f005:**
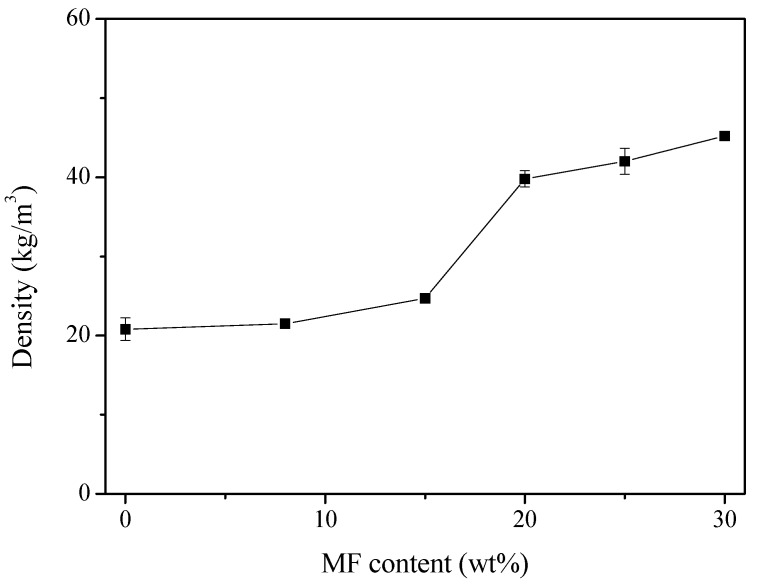
Density of polyurethane /melamine-formaldehyde (PU/MF) foams as a function of MF content, reproduced from [97] with permission, copyright Springer Nature, 2016.

**Figure 6 polymers-11-00953-f006:**
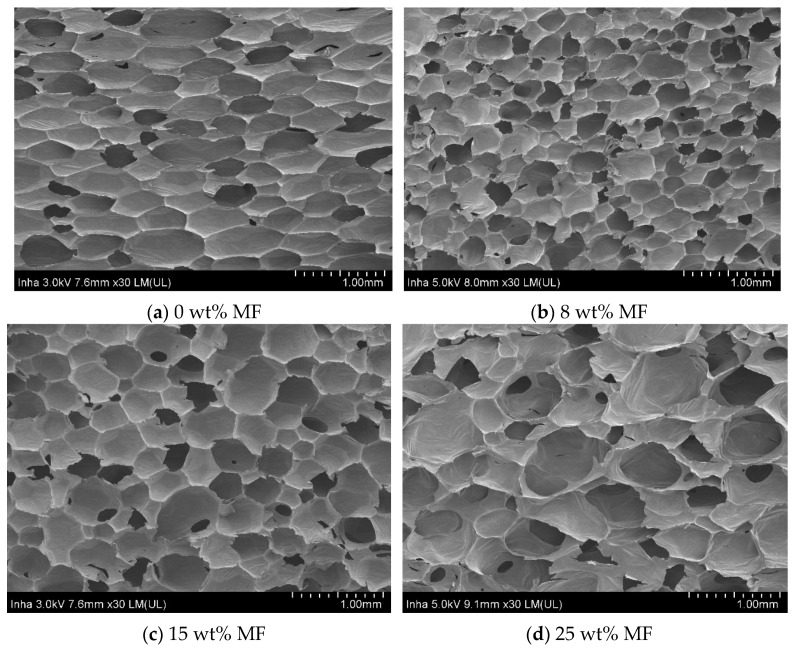
SEM micrographs of PU/MF foams (magnification of 30), reproduced from [97] with permission, copyright Springer Nature, 2016.

**Figure 7 polymers-11-00953-f007:**
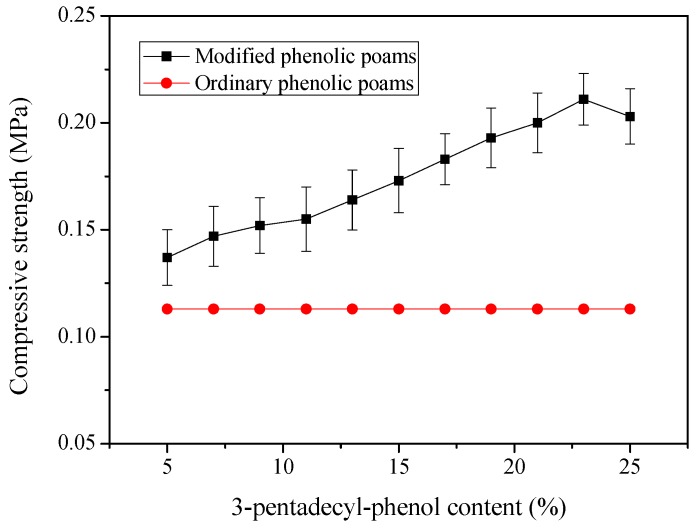
Compressive strength of modified phenolic foams as a function of 3-pentadecyl-phenol, reproduced from [21] under open access license.

**Figure 8 polymers-11-00953-f008:**
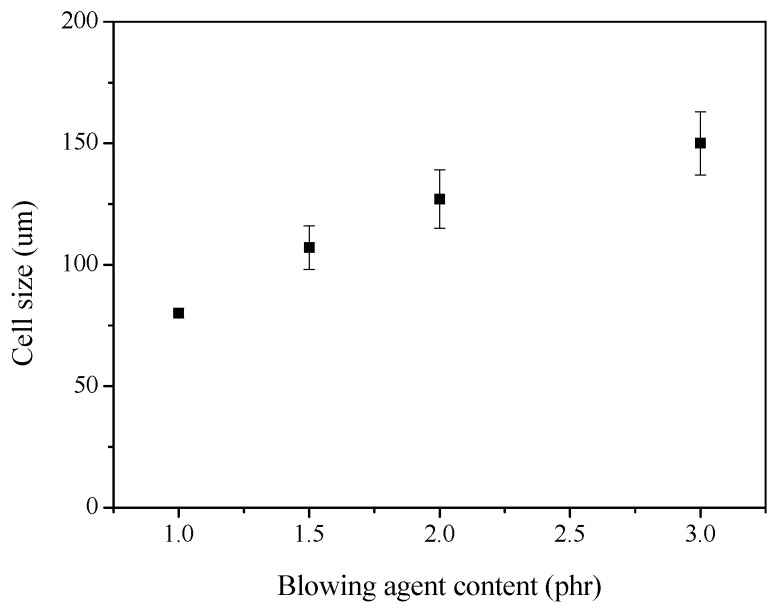
Cell size of polyoxymethylene (POM) foams as a function of blowing agent content, reproduced from [100] with permission, copyright Elsevier, 2016.

**Figure 9 polymers-11-00953-f009:**
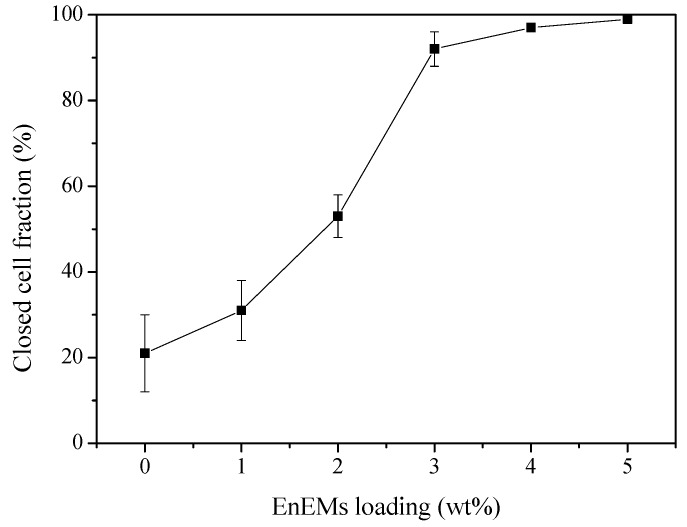
Closed cell fraction of poly(butylene succinate) (PBS) foams as a function of endothermic expandable microsphere (EnEMs)/AC loading ratio, reproduced from [115] with permission, copyright Elsevier, 2018.

**Figure 10 polymers-11-00953-f010:**
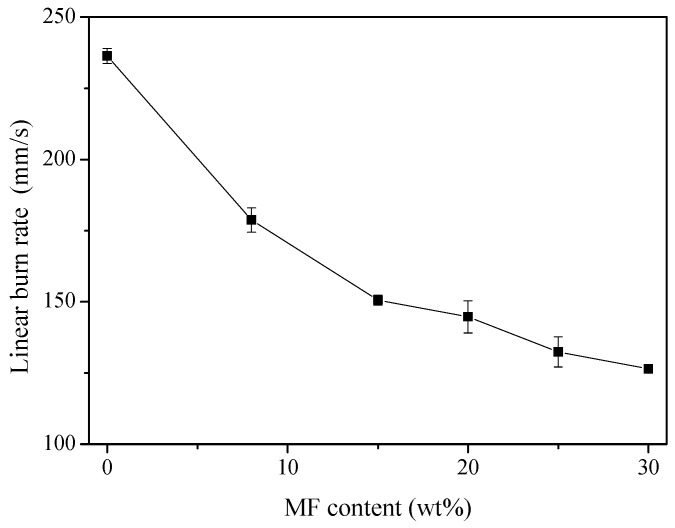
Linear burning rate of PU/MF foams as a function of MF content, reproduced from [97] with permission, copyright Springer Nature, 2016.

**Figure 11 polymers-11-00953-f011:**
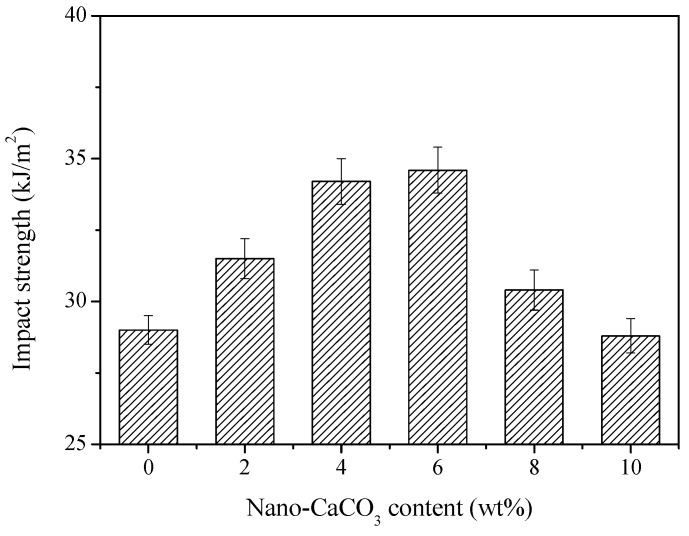
Impact strength of polypropylene (PP) foams as a function of nano-CaCO_3_ content, reproduced from [18] under open access license.

**Table 1 polymers-11-00953-t001:** Density of epoxy foam samples, reproduced from [99] with permission under open access license.

Temperature	160 °C	170 °C	180 °C	190 °C	200 °C
Density (kg⋅m^−3^)	909 ± 14	797 ± 22	676 ± 27	495 ± 38	462 ± 42

**Table 2 polymers-11-00953-t002:** Compressive mechanical properties of PA foams, reproduced from [3] under open access license.

Samples	Density (g/cm^3^)	Compressive Moduli (MPa)	Highest Compressive Stress (MPa)
1	0.096 ± 0.002	1.38 ± 0.15	1.08 ± 0.05
2	0.095 ± 0.002	1.33 ± 0.10	1.23 ± 0.06
3	0.092 ± 0.002	1.29 ± 0.07	1.12 ± 0.03
4	0.090 ± 0.001	1.05 ± 0.06	0.96 ± 0.06
5	0.066 ± <0.001	1.02 ± 0.04	0.92 ± 0.07
6	0.082 ± 0.001	1.14 ± 0.04	1.21 ± 0.04
7	0.101 ± <0.001	4.74 ± 0.08	1.53 ± 0.05

**Table 3 polymers-11-00953-t003:** Mechanical properties of selected soft polyvinyl chloride (PVC) product foamed under 700 W microwave power, reproduced from [56] with permission, copyright Elsevier, 2017.

Material	Tensile Strength (MPa)	Elongation at Break (%)	Apparent Density (kg/m^3^)
PVC + CB + AC	30.8 ± 2.6	3.82 ± 0.41	896 ± 52
PVC + CB + AC + Talc	29.9 ± 1.0	4.21 ± 0.27	892 ± 24
PVC + CB + AC + MMT	29.6 ± 0.4	3.73 ± 0.69	811 ± 34
PVC + CB + AC + TiO_2_	27.5 ± 0.3	3.78 ± 0.22	838 ± 18

Note: CB: Carbon black, AC: Azodicarbonamide, MMT: Montmorillonite.

**Table 4 polymers-11-00953-t004:** Limiting oxygen index (LOI) and UL 94 rating of untreated and toughened phenolic foam, reproduced from [130] with permission, copyright Elsevier, 2018.

Samples	LOI (%)	UL 94 Rating
Untreated phenolic foam	38.0	V0
phenolic foam/1Mg-rGO	38.5	V0
phenolic foam/2Mg-rGO	39.5	V0
phenolic foam/3Mg-rGO	40.5	V0
phenolic foam/4Mg-rGO	41.5	V0
phenolic foam/5Mg-rGO	41.0	V0
phenolic foam/4Mg-AMP	40.0	V0
phenolic foam/4GO	39.5	V0

Note: Mg-AMP: Magnesium amino-tris-(methylenephosphonate) Mg-rGO: Mg-AMP-reduced graphene oxide, GO: Graphene oxide.

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
