# Peer review of "Recent Trends of Foaming in Polymer Processing: A Review"

_polymers, 2019, doi:10.3390/polym11060953_

Reviewer 1 Report

Reviewer’s comments on the manuscript

Recent trends of foaming in polymer processing: a review

            This corrected paper focuses on the main recent articles about polymer foams and foaming process. For biodegradable polymer foams and recycling waste polymer foams, current problems and possible solutions were reported. A considerable effort has been made by the authors to make this paper a review: it is better structured, better illustrated, and more complete than in its first version. As demanded, the conclusion has been completely rewritten.

           However, errors occurred in the listing of references of the corrected version. In particular, the reference [92] (line 129) should have the number [66], which modifies the numbers of references that follow.

Minor remarks:

          Replace [582] by [52], in line 79

          So, I do not recommend acceptance of the paper. Indeed, the modification of the list of references is necessary to consider the publication of the paper.

Author Response

The list of references was modified according to the comments of the reviewer. The reference [92] (page 9) was replaced by [66]. The reference [582] (page 6) was replaced by [52].

Reviewer 2 Report

After the revision, section 4 becomes much better. But I still have concerns about section 3 Research progress. The authors should provide more insights for comparison between different methods. In particular, how are mechanical foaming, physical foaming and chemical foaming compared to each? For each category, what are the advantages and disadvantages of each method that was reviewed? More discussion between different methods are needed, rather than simply describing what each method did. 

Author Response

The advantages and disadvantages of each method were explained according to the comments of the reviewer. The sentences were made in lines 125-128, page 4, lines 135-140, page 4, and lines 149-154, page 5.

The mechanical foaming, physical foaming, and chemical foaming were compared according to the comments of the reviewer. The sentences were made in line 215-226.

This manuscript is a resubmission of an earlier submission. The following is a list of the peer review reports and author responses from that submission.

Round  1

Reviewer 1 Report

This review is intended to discuss recent trends in polymer-based forming technologies. The topic fits the interest of the journal, but the reviewer does not recommend it for publication for the following reasons:

Most contents of the paper, i.e., section 1~6, include very general information on polymer foaming and processing, but they account for almost 2/3 of the paper. Section 7 is focused on “recent progress” and should be extended.

Each section is simply started with something like “…methods can be divided into….” The authors were simply reviewing different classifications of each technique, but did not provide their insights for those technologies.

No further discussion about future research trends is given.

Reviewer 2 Report

Reviewer’s comments on the manuscript

Recent trends of foaming in polymer processing: a review

            This paper is interesting because it is a complete and structured review of the main recent articles about polymer foams and foaming process. For biodegradable polymer foams and recycling waste polymer foams, current problems and possible solutions were reported. But, it seems to me that the conclusion should be more developed, giving the authors’ critical opinion on the literature and their point of view concerning the future advances in the domain of the foams.

Major remark:

In Figure 1, replace “formaion” by “formation”

 Considering these points, I do not recommend acceptance of the paper. Moderate revisions are required, and, in particular, the authors’ personal point of view is expected in the conclusion.  

 Reviewer 3 Report
